# Two-Dimensional Hetero- to Homochiral Phase Transition from Dynamic Adsorption of Barbituric Acid Derivatives

**DOI:** 10.3390/nano13162304

**Published:** 2023-08-10

**Authors:** Fabien Silly, Changzhi Dong, François Maurel, Xiaonan Sun

**Affiliations:** 1TITANS, SPEC, CEA, CNRS, Université Paris-Saclay, 91191 Gif sur Yvette, France; fabien.silly@cea.fr; 2ITODYS, CNRS UMR 7086, Université Paris Cité, 15 rue Jean Antoine de Baïf, 75013 Paris, France

**Keywords:** 2D chiral, heterochiral, homochiral, enantiomer, self-assembly, hydrogen bond, host-guest system

## Abstract

Barbituric acid derivative (TDPT) is an achiral molecule, and its adsorption on a surface results in two opposite enantiomerically oriented motifs, namely TDPT-S_p_ and R_p_. Two types of building blocks can be formed; block I is enantiomer-pure and is built up of the same motifs (format S_p_S_p_ or R_p_R_p_) whereas block II is enantiomer-mixed and composes both motifs (format S_p_R_p_), respectively. The organization of the building blocks determines the formation of different nanoarchitectures which are investigated using scanning tunneling microscopy at a liquid/HOPG interface. Sophisticated, highly symmetric “nanowaves” are first formed from both building blocks I and II and are heterochiral. The “nanowaves” are metastable and evolve stepwisely into more close-packed “nanowires” which are formed from enantiomer-pure building block I and are homochiral. A dynamic hetero- to homochiral transformation and simultaneous multi-scale phase transitions are demonstrated at the single-molecule level. Our work provides novel insights into the control and the origin of chiral assemblies and chiral transitions, revealing the various roles of enantiomeric selection and chiral competition, driving forces, stability and molecular coverage.

## 1. Introduction

Chirality is a natural phenomenon which presents an essential scientific issue in fields ranging from the origin of life to enantioselective/specific catalysis in chemistry and from nonlinear optics to surface nanostructures, etc. Due to the rapidly expanding number of investigations of large molecules with various supramolecular organizations on solid surfaces, two-dimensional (2D) chirality [1] as a phenomenological concept has been increasingly observed and interpreted on the nanoscale. Chiral building blocks can be formed from chiral, prochiral or achiral molecules in the presence of surfaces [2,3,4,5], where chirality can be propagated into long-range nanostructures [6,7,8,9]. Organizational chirality in nanoarchitectures frequently involves 2D mirror symmetry and is produced through the self-assembly of organic molecules (chiral or achiral) or building blocks [6,7,8,9,10,11,12,13,14,15,16]. 

Chiral nanostructures can be directly visualized by means of Scanning Tunneling Microscopy (STM) under different environments in an ultrahigh vacuum (UHV) [4,12,15,16,17,18,19,20] or at a solid/liquid interface [3,5,21,22,23,24,25]. The main scientific and practical interest started with the simple recognition of chirality and then shifted towards the induction and engineering of more complex chiral assemblies [15,16,17,18,19,20,21,22,23,24,25,26,27].

In clean UHV environments, molecules form large enantiomer islands on metal surfaces, and metal–organic interface-directed chiral transfer occurs [28]. Controlled coverage [29,30,31,32] and temperature [33] have served as driving forces for chiral selectivity. Very recently, on-surface reaction has been revealed as a novel means to generate 2D chiral frameworks [34]. Rotation and dynamics of chirality-specific motors, though challenging, have been successfully recorded [35,36]. Although access to different metal surfaces or temperatures at the solid/liquid interface is limited, extra factors such as solvents, molecular concentration or pH can be exploited for chiral propagation and control. For example, chiral assemblies can be induced by mixing an achiral building block with a chiral modifier [21,22,23,37,38]. Solvents can be used to tune 2D chirality [39]. A “chiral memory concept” has been reported [40] where chirality remains even after the chiral modifier has been removed.

On the other hand, it is very important to understand the correlation between the different types of enantiomerically mixed or pure chiralities and chiral-selective phase transitions, but this remains very challenging [29,31,41] and will therefore be the focus of this work. We report a supramolecular self-assembly of prochiral barbituric acid derivatives on a highly oriented pyrolytic graphite (HOPG) substrate. Scanning tunneling microscopy images reveal that the molecules first self-assemble into sophisticated “nanowaves” at the liquid/HOPG interface where the highly symmetric structure is enantiomer-mixed and is heterochiral. The “nanowaves” are metastable and evolve stepwisely into more close-packed “nanowires” which are enantiomer-pure and homochiral. A hetero- to homochiral phase transition is demonstrated at the single-molecule level. The enantioselective competition induces a supramolecular reorganization, which demonstrates the occurrence of multi-scale phase transitions.

## 2. Materials and Methods

5-(4-Tetradecyloxybenzylidene)pyrimidine-2,4,6-trione (TDPT, formula C_25_H_36_N_2_O_4_, Figure 1a) was synthesized according to the known procedure described in previously published articles [42]. 4-Hydroxybenzaldehyde was first treated with NaH in DMF, and the phenolate obtained was condensed with 1-bromotetradecane to provide 1 in excellent yield (95%) and pure enough to be used directly in the next step. Reaction of 1 (Appendix A) with barbituric acid in the presence of a catalytic quantity of piperidinium benzoate gave 2 in good yield (64%).

TDPT was dissolved in 1-phenyloctane and deposited on a highly ordered pyrolytic graphite (HOPG) substrate. Dilute (10^−4^ to 10^−3^ M) and more concentrated (10^−2^ M) solutions were used for the deposition of “nanowaves” or “nanowires”. STM images of the samples at the liquid/solid interface were acquired on a SPM Nanoscope IV (Veeco, Bruker) scanning tunneling microscope. Cut Pt/Ir tips (Goodfellow) were used to obtain constant-current images at room temperature with a bias voltage applied to the sample. STM images were processed and analyzed by means of the FabViewer application [43].

## 3. Results

TDPT is an achiral molecule which does not contain any chiral elements. Its adsorption on a surface results in two opposite enantiomerically oriented motifs indicating that the molecule is prochiral [15] and that the surface plane serves as a 2D mirror (orange dashed line in Figure 1b). The two enantiomeric motifs are defined by treating the surface as a substituent to a tetrahedral center and then applying the Cahn–Ingold–Prelog priority rules [44]. The surface is considered to have the highest priority (3), and the hydrogen (of the CH bridging the two rings) is considered to have the lowest (0). The barbituric ring has priority (2), and the phenyl ring has priority (1). As shown in Appendix A (see Appendix A), the configuration on the left can then be defined as “S_p_” and that on the right as “R_p_” where p means planar (S_p_ and R_p_ in red and green, respectively; Figure 2b).

Different building blocks can be formed from the two enantiomeric motifs through intermolecular hydrogen bonding. Building block I, composed of two identical enantiomers (S_p_S_p_ or R_p_R_p_, Figure 1c, left), is homochiral. On the contrary, building block II, made up of two opposite enantiomers (S_p_R_p_, Figure 1c, right), is heterochiral. In the case of S_p_S_p_ or R_p_R_p_ building blocks, the same enantiomeric motifs interact with one strong N-H**∙∙∙**O and one weak C-H**∙∙∙**O hydrogen bond. In the S_p_R_p_ blocks, the opposite motifs interact with two strong N-H**∙∙∙**O and one weak C-H**∙∙∙**O hydrogen bonds. The supramolecular organization from these different building blocks I and II can result in different nanoarchitectures and will be investigated under high-resolution STM at the solid/liquid interface.

The large-scale STM image reveals that TDPT molecules self-assemble into highly ordered 2D nanoarchitectures at the 1-phenyloctane/HOPG interface at room temperature. Dilute solutions with concentrations between 10^−4^ and 10^−3^ M are used for deposition. Very complex supramolecular wave-shaped nanopatterns are clearly visible (Figure 2a). Within the “nanowaves”, adjacent arcs curve in the same way. Two facing arcs from neighboring waves show almost perfect 2D mirror symmetry with respect to the surface plane. In the center of symmetry of two facing arcs (curving oppositely), nanosized cavities are formed where two guest TDPT molecules are nested in a guest–host system. The green circle (Figure 2b), superimposed on the STM image in Figure 2b, highlights the position of paired guest molecules in the host structure. The “head” of each TDPT molecule appears as two bright spots in the high-resolution STM image (Figure 2b) due to a higher electron density of states (DOS) whereas the alkyl chain appears darker. To form the double-line “nanowaves”, TDPT molecules are oriented in head-to-head fashion, perpendicular to the aligned axes of the wave patterns. The network unit cell (Figure 2b, red dashed lines) is a parallelogram with cell constants of 4.0 nm (a), 6.3 nm (b) and an angle of around 110°. The unit cell is surprisingly large, composed of 14 molecules, i.e., 12 molecules forming the host structure and 2 guest molecules. The alkyl chains of the TDPT molecules (guest and host molecules) are aligned in parallel to each other, which maximizes van de Waals (VDW) interactions.

A supramolecular model of the TDPT “nanowave” structure is proposed in Figure 2c. The host unit cell is constructed from 8 S_p_ and 4 R_p_ molecules (the unit cell can also contain 4 S_p_ and 8 R_p_ molecules in other domains to be enantiomerically equal), where TDPT-S_p_ and TDPT-R_p_ are the two motifs enantiomerically opposite (Figure 1b). Two guest molecules occupy each cavity, and they can be either S_p_S_p_ or R_p_R_p_ blocks. Moreover, the homochiral building block I (S_p_S_p_/R_p_R_p_, Figure 1c) and heterochiral building block II (S_p_R_p_, Figure 1c) both contribute with an enantiomeric S_p_/R_p_ (or R_p_/S_p_) ratio of 2:1 inside one host unit cell (without considering the guest molecules). The “nanowave”, with its more symmetrical structure, is a 2D heterochiral nanoarchitecture. The STM images show that the smallest separation between the two neighboring “nanowaves” is around 2δ = 3.3 nm. This value is twice the length of the C_14_H_29_ alkyl chains (1.7 nm), which reveals that there is no chain intercrossing (Figure 2c) from neighboring “nanowaves”. The detailed symmetry analysis has been published elsewhere [14]. In this structure, molecules interact head-to-head by N-H∙∙∙O hydrogen bonds and side-to-side by O-H∙∙∙O and N-H∙∙∙O hydrogen bonds (model in Figure 2c). Intermolecular H-bonding and molecule/substrate VDW interactions stabilize the “nanowaves”.

The TDPT “nanowaves” are formed immediately after deposition, as previously demonstrated (Figure 2). A few hours after deposition, short nano-length wires (green dashed circles in Figure 3a,b) appear between the “nanowaves” at the initial guest positions. The nanowires are doubled (Figure 3) with molecules organized in head-to-head fashion. The short molecular wires continue to grow. A transition area is recorded (pink arrow in Figure 3a,b) where the host “nanowaves” are pushed apart and their original curved arc symmetry is lost.

A molecular model is proposed to interpret the initial formation of the nanowires (Figure 3c). In contrast to the “nanowaves” which are based on the alternation of S_p_ and R_p_ enantiomeric motifs with a ratio of 2:1 (composed of building blocks I and II), the short wire is formed from mono-enantiomeric S_p_S_p_ or R_p_R_p_ (building blocks I only). In a transition area between the “nanowave” and the “nanowire” (yellow dashed square in Figure 3c), S_p_ and R_p_ motifs alternate with a ratio of 1:1 (building block II only). The dynamic evolvement of the short nanowires causes local symmetry breaking within the initial “nanowave” organizations and is considered as an early stage of a structural as well as chiral phase transition, where the “nanowave” is heterochiral, and the nanowire is homochiral.

When the initial “nanowave” structure is left to evolve on the surface long enough (tens of hours), the short nanowires continue to deposit on the surface and turn into longer wires replacing the “nanowaves”. Eventually, the surface organization is overwhelmed by almost linear long wires. The high-resolution STM image (Figure 4b) reveals that molecules are arranged in head-to-head fashion, making the wires double, whereas the alkyl chains are in parallel perpendicular to the main axes of the wires. The unit cell is marked by the red rectangle with cell constants of around 5.6 (a) nm and 1.6 (b) nm and an angle close to 90°. The unit cell of the network is composed of 4 TDPT molecules. The same long nanowire structure can be directly obtained by deposition of a more concentrated TDPT solution (10^−2^ M). This indicates that wires are preferred at high molecular coverage.

The proposed model (Figure 4c) suggests that the double-wire structure is constructed mainly from mono-enantiomeric building block I (either S_p_S_p_ or R_p_R_p_) making the structure homochiral. The same enantiomeric motifs are stabilized head-to-head and side-to-side through N-H∙∙∙O and C-H∙∙∙O hydrogen bonds. Occasionally, an enantiomer-mixed building block II is inserted into the homochiral wires as a defect which generates local bending of the wire (Figure 4c). It is possible that the different long-wire domains are built up from either S_p_S_p_ or R_p_R_p_ chiral building block I so that the numbers of the two enantiomeric motifs are the same on the whole surface. However, the long-wire structure, with an enantiomer-pure unit cell, is reasonably considered as homochiral. Different domains of the long wires coexist with orientation angles around 60° or 120° induced by the three-fold symmetry of the substrate (Figure 4a). The separation between the two neighboring TDPT wire is δ = 1.7 nm which corresponds to the length of the C_14_H_29_ alkyl chain. The alkyl chains from adjacent wires therefore intercalate, making the structure close-packed and closing any cavities.

Molecular dynamics on surfaces is an essential point in nano-engineering to control self-assemblies. The investigation or recording of such dynamics remain very challenging, depending on the time-scale match between the STM imaging and the on-surface evolvement. In our system, dynamic adsorption of the TDPT molecules can be visualized through a series of scans. A real-time movie (from continuous STM images) is recorded (Appendix A file) providing an overview of homochiral TDPT wire deposition on a time-scale of around 20 min; Figure 5a–c show stills from the movie. We demonstrate that the TDPT wires continue to grow locally with time (indicated by the pink, blue and green circles in Figure 5a–c) at the liquid/HOPG interface until the surface is completely covered, and finally, there are no more cavities. As we have understood from the previous discussion that the alkyl chains of the TDPT molecules are aligned in parallel all along the HOPG [100] direction, the molecular wires therefore grow along the HOPG [1110] direction. Each TDPT molecule adsorbed is attached to its neighboring molecules side-to-side and head-to-head through hydrogen bonds and VDW interactions. Both intermolecular and molecule-substrate interactions obviously play important roles in the continuous molecular adsorption on the surface. Different domains are observed to have angles around 60° with respect to each other, which is induced by the HOPG substrate. Neither the STM scan nor the applied bias voltage influences or disturbs molecular wire growth. This phenomenon indicates that the close-packed, long wires are highly favored structures on-surface, and that the “nanowave” to “wire” structural phase transition is irreversible.

## 4. Discussion and Conclusions

Once adsorbed on a surface, both enantiomeric motifs are created from the prochiral TDPT molecules. The S_p_ and R_p_ enantiomeric motifs form hetero- and homochiral building blocks I and II, respectively. The homochiral and heterochiral interactions compete in the formation of the nanoarchitectures. “Nanowaves” and “nanowires” are generated, and a transition area between the two structures is attributed to different organizations of the two building blocks. Briefly, the “nanowave” (Figure 6a) is composed of both I and II building blocks; it is a metastable host structure (with nanocavities). The unit cell is enantiomer-mixed with an S_p_/R_p_ (or R_p_/S_p_) ratio of 1:2. The “transition area” (Figure 6b) is made up of building block II (S_p_/R_p_ ratio 1:1); it is unstable and appears only when the nanowave structural symmetry breaks due to growth of the initial nanowires. The nanowire (Figure 6c) consists of building block I only, and the unit cell, composed of purely S_p_ or R_p_ motifs, is an enantiomer-separated structure. The nanowave’s unit cell is heterochiral whereas the wire’s is homochiral.

The “nanowaves” and the “nanowires” are both stabilized through similar hydrogen bonds as well as molecule/substrate interactions. Moreover, in the nanowire structures, side-chains intercalate in parallel (Figure 6c) [45,46,47,48], and an extra VDW force is involved. This extra VDW between parallel chains is chiro-selective and exists only in homochiral organizations. The competition between hetero- and homochirality, as well as the different chiro-selective intermolecular forces, plays an essential role in the structural transitions. The homochiral “nanowires” are the more favored on-surface structures. Eventually, the two TDPT enantiomers should appear in equal numbers for the surface organizations to remain globally achiral.

To summarize, we observe a dynamic multiphase transition of TDPT molecules: first, a structural phase transition occurs where “nanowaves” are transformed into “nanowires”; second, this involves a hetero- to homochiral transition; third, during the transition process, chiro-selective intermolecular interactions play essential roles. The “nanowave” is stabilized by both molecular/substrate interactions and intermolecular hydrogen bonds. The “nanowires” are stabilized by similar forces, but an extra intermolecular VDW force from intercalated alkyl chains is involved, and this favors the homochiral structure. The irreversible dynamic deposition of wires is another proof of their higher stability. Moreover, the “nanowave” is a guest–host system, and the “nanowires” are more closely packed and have a higher surface coverage. Our work provides novel insights into the control and the origin of chiral assemblies and chiral transitions at the molecular scale, revealing the various roles of enantiomeric selection and chiral competition, driving forces, stability and molecular coverage.

## Figures and Tables

**Figure 1 nanomaterials-13-02304-f001:**
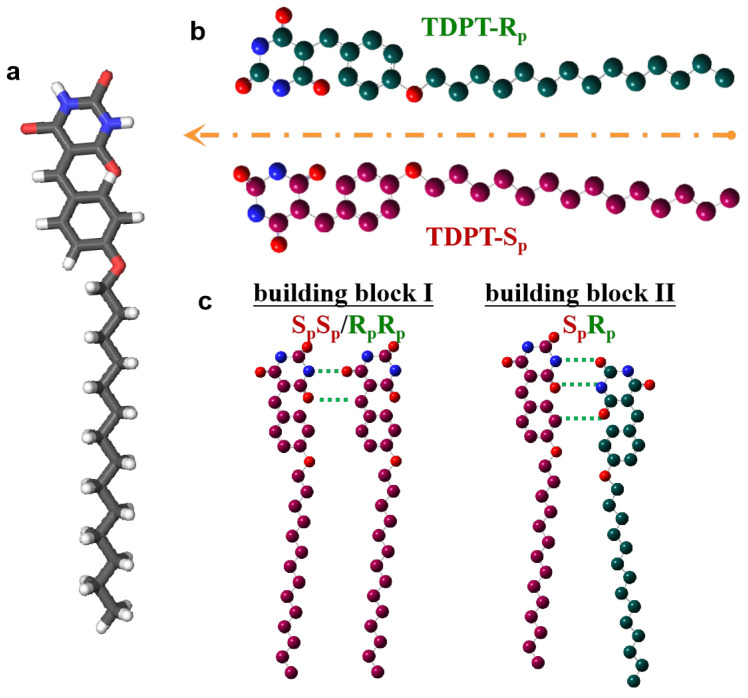
Molecular models: (**a**) TDPT molecular structure where C, N, O and H elements are grey, dark blue, red and white balls, respectively. (**b**) Surface-supported TDPT-S_p_ and R_p_ enantiomeric motifs showing 2D mirror symmetry. (**c**) Building blocks I (homochiral) and II (heterochiral) which are made from the same (S_p_S_p_/R_p_R_p_) and opposite (S_p_R_p_) enantiomers, respectively. The green dashed lines indicate the possible formation of hydrogen bonds.

**Figure 2 nanomaterials-13-02304-f002:**
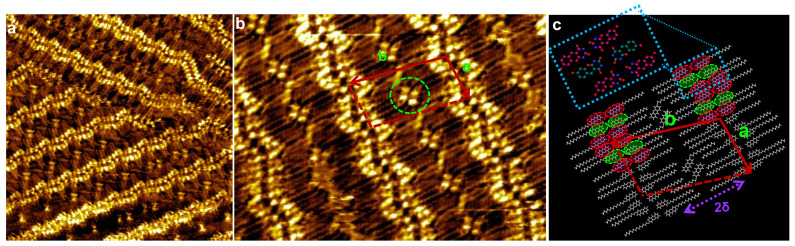
(**a**,**b**) Large-scale (44 × 40 nm^2^) and high-resolution (20 × 14 nm^2^) STM images of double-line “nanowaves” which are also guest–host systems. I_t_ = 30~60 pA, U_s_ = 0.3~0.5 V. (**c**) Molecular modeling shows the structural organization. Red and green circles indicate the different S_p_ or R_p_ enantiomeric motifs. Inset model shows that neighboring molecules are organized in mixed S_p_S_p_/R_p_R_p_ and S_p_R_p_ modes.

**Figure 3 nanomaterials-13-02304-f003:**
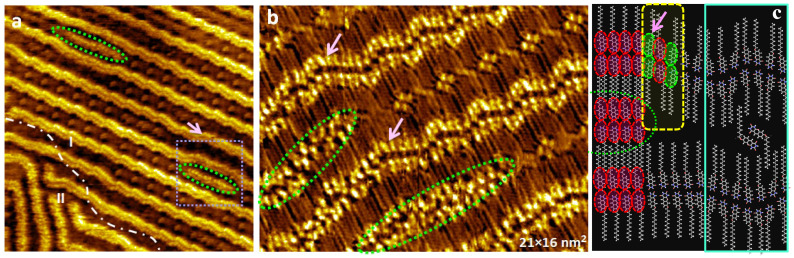
(**a**,**b**) Large-scale (55 × 55 nm^2^) and high-resolution (21 × 16 nm^2^) STM images. “Nanowires” start to grow between the “nanowaves”. I_t_ = 30~60 pA, U_s_ = −0.3~−0.5 V. (**c**) Molecular modeling shows the transition between the two structures. Red and green circles indicate the different S or R enantiomers. Yellow rectangle highlights the transition area corresponding to the pink arrow in image (**b**).

**Figure 4 nanomaterials-13-02304-f004:**
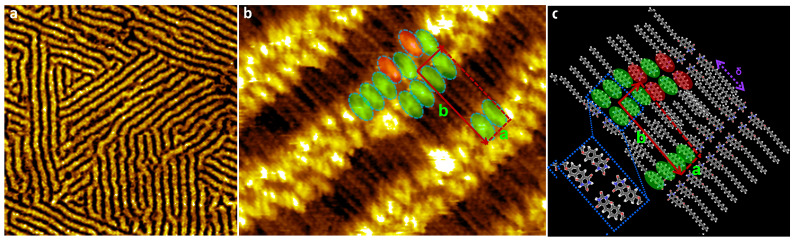
STM images of TDPT molecular wires on HOPG. (**a**) 75 × 75 nm^2^, I_t_ = 30 pA, U_s_ = −0.4 V. (**b**) 14 × 7 nm^2^, I_t_ = 40 pA, U_s_ = −0.3 V. The red rectangle highlights the unit cell of the structure. (**c**) Molecular model of the double wires. The green and red circles indicate the R_p_ and S_p_ enantiomeric motifs, respectively. The purple dashed arrow shows the separation between the two wires (δ~1.7 nm).

**Figure 5 nanomaterials-13-02304-f005:**
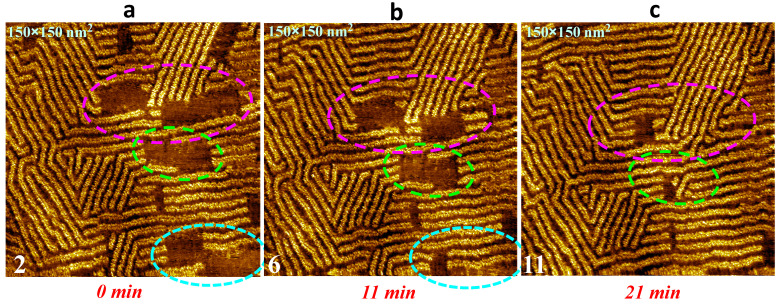
Real-time STM stills of TDPT wires being continuously deposited on HOPG surface in a 21 min time window. The different pink, green and blue circles indicate where the wires fill the surface cavities: 150 × 150 nm^2^, I_t_ = 30 pA, U_s_ = −0.4 V.

**Figure 6 nanomaterials-13-02304-f006:**
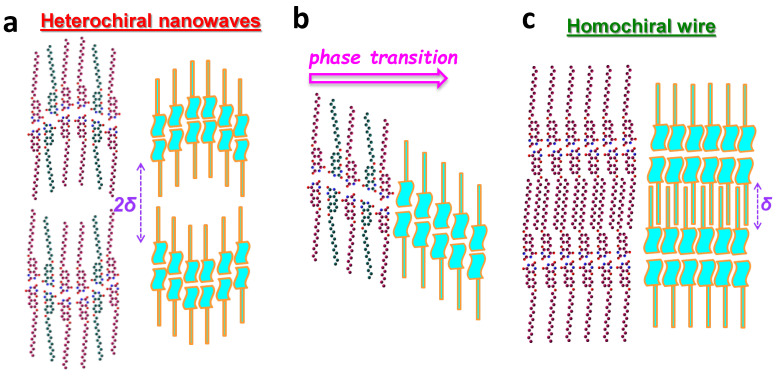
Schematic illustration of: (**a**) Heterochiral nanowave, (**b**) Transition area, and (**c**) Homochiral wires.

## Data Availability

Not applicable.

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
