# Peer review of "Two-Dimensional Hetero- to Homochiral Phase Transition from Dynamic Adsorption of Barbituric Acid Derivatives"

_nanomaterials, 2023, doi:10.3390/nano13162304_

Round 1
Reviewer 1 Report
The Manuscript “2D Hetero- to Homo- Chiral Phase Transition from Dynamic Absorption of Barbituric Acid Derivatives” by F. Silly, C. Dong, F. Maurel and X. Sun describes dynamic hetero- to homochiral transformation of a self-assembled surface aggregates formed by rigid molecules of achiral 5-benzylidene derivative of barbiruric acid. Self-organization of small molecules on metallic and non-metallic surfaces is a unique phenomenon, which can give rise to local chiral motifs and even to directed modification of chiral motifs. Chiral adsorption motif originated from achiral molecules can further be applied in chiral recognition and constructing of more complex homo- and heterochiral patterns. Studies within this fast-evolving part of modern nanotechnology are urgent and contribute significantly to understanding the origins of chiral assemblies and development of novel materials.
There are several terminological issues that have to be revised prior the manuscript can be published.
Terms “enantiomer” and “chiral” are interpreted freely throughout the manuscript. For example, it is postulated that “prochiral” molecules of barbituric acid derivatives “become chiral once absorbed on surface”. I would like to emphasize that molecule of compound TDPT is ACHIRAL (it does not contain any chiral elements, neither centers nor planes; it is superposable with its mirror image). Only the adsorption motif can be considered as a chiral one.
The statement that molecules of TDPT lying on their opposite sides are enantiomeric is erroneous. These molecules are identical and superposable with each other! Though adsorption motifs formed by molecules lying on one side or other side are chiral indeed. Turning the TDPT molecules within homochiral motif from their one side to the other side eventually results in the reversal of configuration of adsorption motif. These aspects and probable mechanisms of hetero- to homochiral phase transition should be mentioned within the text.
It is unclear how authors assigned R and S stereodiscriptors to TDPT molecules turned to the surface by various sides (Fig. 1b) or building blocks I and II (Fig. 1c). I suggest not to apply R/S nomenclature to an assembly of achiral molecules in a chiral manner. Probably, some kind of nomenclature should be introduced. Anyway, the designation of configuration should be discussed in the text.
I believe that the article by F. Silly, C. Dong, F. Maurel and X. Sun will be interesting for the broad auditory of the Nanomaterials journal and recommend publishing it after revision of stereochemical nomenclature discussed above.
Author Response
Please find attached to this letter answers to the reviewers comments of our manuscript entitled “2D Hetero- to Homo- Chiral Phase Transition from Dynamic Absorption of Barbituric Acid Derivatives” that we wish to publish in Nanomaterials. Its ID is nanomaterials-2540889. The reviewer comments have been taken into account as follows:
Reviewer1:
The Manuscript “2D Hetero- to Homo- Chiral Phase Transition from Dynamic Absorption of Barbituric Acid Derivatives” by F. Silly, C. Dong, F. Maurel and X. Sun describes dynamic hetero- to homochiral transformation of a self-assembled surface aggregates formed by rigid molecules of achiral 5-benzylidene derivative of barbiruric acid. Self-organization of small molecules on metallic and non-metallic surfaces is a unique phenomenon, which can give rise to local chiral motifs and even to directed modification of chiral motifs. Chiral adsorption motif originated from achiral molecules can further be applied in chiral recognition and constructing of more complex homo- and heterochiral patterns. Studies within this fast-evolving part of modern nanotechnology are urgent and contribute significantly to understanding the origins of chiral assemblies and development of novel materials.
There are several terminological issues that have to be revised prior the manuscript can be published.
- Terms “enantiomer” and “chiral” are interpreted freely throughout the manuscript. For example, it is postulated that “prochiral” molecules of barbituric acid derivatives “become chiral once absorbed on surface”. I would like to emphasize that molecule of compound TDPT is ACHIRAL (it does not contain any chiral elements, neither centers nor planes; it is superposable with its mirror image). Only the adsorption motif can be considered as a chiral one.
Author’s reply: We thank the reviewer for the comments and suggestions. We agree that we should state that the TDPT molecule itself is achiral. We have now added one sentence “TDPT is an achiral molecule which does not contain any chiral elements.” At the beginning of page 3.
- The statement that molecules of TDPT lying on their opposite sides are enantiomeric is erroneous. These molecules are identical and superposable with each other! Though adsorption motifs formed by molecules lying on one side or other side are chiral indeed. Turning the TDPT molecules within homochiral motif from their one side to the other side eventually results in the reversal of configuration of adsorption motif. These aspects and probable mechanisms of hetero- to homochiral phase transition should be mentioned within the text.
Author’s reply: The reviewer is right. To be more precise, we now change the statement as follows: “The adsorption of TDPT on a surface results in two opposite enantiomeric motifs indicating that the molecule is prochiral and that the surface plane serves as a 2D mirror (orange dashed line in Figure 1b). …”
- It is unclear how authors assigned R and S stereodiscriptors to TDPT molecules turned to the surface by various sides (Fig. 1b) or building blocks I and II (Fig. 1c). I suggest not to apply R/S nomenclature to an assembly of achiral molecules in a chiral manner. Probably, some kind of nomenclature should be introduced. Anyway, the designation of configuration should be discussed in the text.
Author’s reply: We agree that the definition of the two enantiomers is not clear. The reviewer 2 has also given a similar comment and has suggested us to use R(p) and S(p) where p means planer. Meanwhile, we have taken some principles from related literature and have discussed with a stereochemist. To make the S and R definition more clear, we have added the following description in the main text page 3 (above Figure 2) “The two enantiomers are defined by treating the surface as a substitutent to a tetrahedral center, and then applying the Cahn-Ingold-Prelog priority rules. The surface is considered to have the highest priority (3), and the hydrogen (of the CH bridging the two rings) the lowest (0). The barbituric ring has priority (2) and the phenyl ring (1). As shown in scheme S1 (see SI), the left configuration can then be defined as “S(p)” and right “R(p)” where p means planar. For the sake of simplicity, in what follows the two enantiomers will be referred to as TDPT-S and TDPT-R (in red and green, respectively; in Figure 2b).”
And the following scheme is added in the SI file :
I believe that the article by F. Silly, C. Dong, F. Maurel and X. Sun will be interesting for the broad auditory of the Nanomaterials journal and recommend publishing it after revision of stereochemical nomenclature discussed above.
Open Review
(x) I would not like to sign my review report
( ) I would like to sign my review report
Quality of English Language
( ) I am not qualified to assess the quality of English in this paper
( ) English very difficult to understand/incomprehensible
( ) Extensive editing of English language required
( ) Moderate editing of English language required
( ) Minor editing of English language required
(x) English language fine. No issues detected
|
Yes |
Can be improved |
Must be improved |
Not applicable |
|
|
Does the introduction provide sufficient background and include all relevant references? |
(x) |
( ) |
( ) |
( ) |
|
Are all the cited references relevant to the research? |
(x) |
( ) |
( ) |
( ) |
|
Is the research design appropriate? |
(x) |
( ) |
( ) |
( ) |
|
Are the methods adequately described? |
( ) |
(x) |
( ) |
( ) |
|
Are the results clearly presented? |
( ) |
( ) |
(x) |
( ) |
|
Are the conclusions supported by the results? |
( ) |
(x) |
( ) |
( ) |
We hope that our revised manuscript will meet the high standard of quality needed for publication in the journal of Nanomaterials.
Thanking you for your consideration of this work.
Sincerely yours,
Xiaonan Sun
University of Paris Diderot
sun.xiaonan@univ-paris-diderot.fr
tel: 33 1 572772 13
http://www.itodys.univ-paris7.fr/fr/

Reviewer 2 Report
The manuscript is based on the previous results of the team (ref. 14) the subject of which is the self-assembly of 5-(4-tetradecyloxybenzylidene)pyrimidine-2,4,6-trione (TDPT) at at the interface of 1-phenyloctane and HOPG. The novelty of the present manuscript lies in the dynamic transition of these assemblies from heterochiral to homochiral entities observed by STM.
The authors should comment on the fact that in the SS/RR assemblies there are one strong N-H...O and one weak C-H...O hydrogen bonds while in the SR assemblies there are two strong N-H...O and one weak C-H...O hydrogen bonds (cf. Fig. 1). How are these strong hydrogen bonds outcompeted by the weak van der Waals interactions of the interdigitating side chains? Computational support would be needed to substantiate these effects. These interdigitating side chains in self-assembly are known in the literature (e.g. M. Dong et al. “Cooperating Dipole-Dipole and Van Der Waals Interactions Driven 2D Self-Assembly of Fluorenone Derivatives: Ester Chain Length Effect.” Physical chemistry chemical physics PCCP 19, no. 46 (2017): 31113–20. https://doi.org/10.1039/c7cp06462d; R. Colle et al. “Structure and X-Ray Spectrum of Crystalline Poly(3-Hexylthiophene) From DFT-Van Der Waals Calculations.” Physica Status Solidi B 248, no. 6 (2011): 1360–68. https://doi.org/10.1002/pssb.201046429; A. Ciesielski et al. “Self-Assembly of N3-Substituted Xanthines in the Solid State and at the Solid–Liquid Interface.” Langmuir 29 (2013): 7283–90. https://doi.org/10.1021/la304540b; A. Di Pierro et al. “Molecular Junctions for Thermal Transport Between Graphene Nanoribbons: Covalent Bonding Vs. Interdigitated Chains.” Computational Materials Science 142 (2018): 255–60. https://doi.org/10.1016/j.commatsci.2017.10.019).
The identification of novel architectures with the names R and S is questionable, these stereodescriptors are reserved for stereogenic centers that can be classified using the Cahn-Ingold-Prelog rules (cf. G. P. Moss. “Basic Terminology of Stereochemistry (IUPAC Recommendations 1996).” Pure and Applied Chemistry 68, no. 12 (1996): 2193–2222). Chirality at surfaces is closest to planar chiralty and the sense of chirality of the mentioned assemblies may be characterised by the stereodescriptors used for planarly chiral entities, R(P) and S(P) (P being subscripted) or P and M.
The manuscript has been prepared carelessly (numerous typos etc. are present), thus it requires extensive editing.
Line 14: "fromboth" should be replaced by "from both".
Line 91: the empirical formula should be written with subscripted numbers.
Line 93: what is [REF]?
Line 95: what is [14 Karan]? Reference 15?
Elemental analyses and/or HR-MS data are missing for compound 2 (Supporting info, pg. 3).
A scale bars are missing from STM images (Supporting info, pg. 4-5).
"raal-time" (Supporting info, pg. 5, Caption to Fig. S3) should be replaced by "real-time".
Author Response
Dear Editor Zhang,
Please find attached to this letter answers to the reviewers comments of our manuscript entitled “2D Hetero- to Homo- Chiral Phase Transition from Dynamic Absorption of Barbituric Acid Derivatives” that we wish to publish in Nanomaterials. Its ID is nanomaterials-2540889. The reviewer comments have been taken into account as follows:
Reviewer: 2
The manuscript is based on the previous results of the team (ref. 14) the subject of which is the self-assembly of 5-(4-tetradecyloxybenzylidene)pyrimidine-2,4,6-trione (TDPT) at at the interface of 1-phenyloctane and HOPG. The novelty of the present manuscript lies in the dynamic transition of these assemblies from heterochiral to homochiral entities observed by STM.
The authors should comment on the fact that in the SS/RR assemblies there are one strong N-H...O and one weak C-H...O hydrogen bonds while in the SR assemblies there are two strong N-H...O and one weak C-H...O hydrogen bonds (cf. Fig. 1). How are these strong hydrogen bonds outcompeted by the weak van der Waals interactions of the interdigitating side chains? Computational support would be needed to substantiate these effects.
Author’s reply: We thank the reviewer for the comments. To make more clear the intermolecular interactions from the different building blocks, we have added in the main text page 3 the following sentence “In the case of SS/RR builiding blocks, the same enantiomers interaction with one strong N-H...O and one weak C-H...O hydrogen bonds. While in the SR blocks, the opposite enantiomers interact with two strong N-H...O and one weak C-H...O hydrogen bonds”.
The competition between these intermolecular hydrogen bonds and VDW interactions is a very complicated procedure. As the reviewer knows, it is quite difficult to simply clarify which kind of interaction is playing what kind of role in different structures. The only thing we can be sure according to our observations is that, the close-packed homochiral structure is stabilized with different hydrogen bonds but also has extra van der Waals interactions thanks to the interdigitating side chains. It seems to us that this extra VDW, which the heterochiral structure did not contain, makes the homochiral structure preferred on surface. This point is stated in the last paragraph of page 7 : “The “nanowaves” and the “nanowires” are both stabilized through similar hydrogen bonds as well as molecule/substrate interactions. Moreover, in the nanowire structures, side-chains intercalate in parallel (Figure 6c), and an extra VDW force is involved. This extra VDW between parallel chains is chiro-selective and exists only in homochiral organizations. The competition between hetero- and homochirality, as well as the different chiro-selective intermolecular forces, plays an essential role in the structural transitions. The homochiral “nanowires” are the more favored on-surface structures.”
Because of the large unit cell of the “nanowave” structure and the long alkyl chains, especially the dynamic transition between the nanowaves and the wire, enormous amount of work will be required for a good calculation using DFT or molecular dynamic simulations. We are having discussions to perform some calculation by the theoreticians, as it will be time consuming, we will probably write a following article mainly focused on the calculation and the interpretation of the different intermolecular competition. We wish the reviewer could kindly understand this difficulty and that the calculation is beyond the scope of of this work.
On another hand, as the reviewer has also mentioned (in the next question below) that some simulation as well as experimental works concerning interdigitating alkyl chain interaction have been reported. We will cite all the references recommended by the reviewer.
- These interdigitating side chains in self-assembly are known in the literature (e.g. M. Dong et al. “Cooperating Dipole-Dipole and Van Der Waals Interactions Driven 2D Self-Assembly of Fluorenone Derivatives: Ester Chain Length Effect.” Physical chemistry chemical physics PCCP 19, no. 46 (2017): 31113–20. https://doi.org/10.1039/c7cp06462d; R. Colle et al. “Structure and X-Ray Spectrum of Crystalline Poly(3-Hexylthiophene) From DFT-Van Der Waals Calculations.” Physica Status Solidi B 248, no. 6 (2011): 1360–68. https://doi.org/10.1002/pssb.201046429; A. Ciesielski et al. “Self-Assembly of N3-Substituted Xanthines in the Solid State and at the Solid–Liquid Interface.” Langmuir 29 (2013): 7283–90. https://doi.org/10.1021/la304540b; A. Di Pierro et al. “Molecular Junctions for Thermal Transport Between Graphene Nanoribbons: Covalent Bonding Vs. Interdigitated Chains.” Computational Materials Science 142 (2018): 255–60. https://doi.org/10.1016/j.commatsci.2017.10.019).
Author’s reply: We thank the reviewer for introducing these references. We have now added these references as following:
Ref 45: Phys. Chem. Chem. Phys., 2017, 19, 31113-31120.
Ref 46: Physica Status Solidi B 248, no. 6 (2011): 1360–68
Ref 47: Langmuir 29 (2013): 7283–90.
Ref 48: Computational Materials Science 142 (2018): 255–60.
3.The identification of novel architectures with the names R and S is questionable, these stereodescriptors are reserved for stereogenic centers that can be classified using the Cahn-Ingold-Prelog rules (cf. G. P. Moss. “Basic Terminology of Stereochemistry (IUPAC Recommendations 1996).” Pure and Applied Chemistry 68, no. 12 (1996): 2193–2222). Chirality at surfaces is closest to planar chiralty and the sense of chirality of the mentioned assemblies may be characterised by the stereodescriptors used for planarly chiral entities, R(P) and S(P) (P being subscripted) or P and M.
Author’s reply: We agree that the definition of the two enantiomers is not clear. We have discussed with stereochemist and therefore have added the following description in the main text page 3 (above Figure 2) “The two enantiomers are defined by treating the surface as a substitutent to a tetrahedral center, and then applying the Cahn-Ingold-Prelog priority rules. The surface is considered to have the highest priority (3), and the hydrogen (of the CH bridging the two rings) the lowest (0). The barbituric ring has priority (2) and the phenyl ring (1). As shown in scheme S1 (see SI), the left configuration can then be defined as “S(p)” and right “R(p)” where p means planar. For the sake of simplicity, in what follows the two enantiomers will be referred to as TDPT-S and TDPT-R (in red and green, respectively; in Figure 2b).”
Moreover, w have discussed with a British native colleague, who suggested to use S and R instead of Sp and Rp, where the later is making the English clumsy.
The reference is now cited in the article as No. 43:
And the following scheme S1 is added in the SI file :
Cmments on the Quality of English Language
The manuscript has been prepared carelessly (numerous typos etc. are present), thus it requires extensive editing.
Line 14: "fromboth" should be replaced by "from both".
We thank the reviewer for the careful checking and we sincerely apologize that we had made these mistakes.
The mistake in the abstract has been corrected where a blank is added between the two words “from both”
Line 91: the empirical formula should be written with subscripted numbers.
The chemical formula is now corrected as “C25H36N2O4” in page 3
Line 93: what is [REF]?
The mistake is corrected and referece No. 42 is added.
Line 95: what is [14 Karan]? Reference 15?
We apologize that we had made this mistake, it is listed in the main text as referece 15.
A scale bars are missing from STM images (Supporting info, pg. 4-5).
Scale bars are added in all STM images in Figure S1 and S3.
"raal-time" (Supporting info, pg. 5, Caption to Fig. S3) should be replaced by "real-time".
This mistake in the SI file has been corrected.
The main manuscript including the references have been re-edited.
Open Review
(x) I would not like to sign my review report
( ) I would like to sign my review report
Quality of English Language
( ) I am not qualified to assess the quality of English in this paper
( ) English very difficult to understand/incomprehensible
( ) Extensive editing of English language required
(x) Moderate editing of English language required
( ) Minor editing of English language required
( ) English language fine. No issues detected
|
|
|
|
Yes |
Can be improved |
Must be improved |
Not applicable |
|
Does the introduction provide sufficient background and include all relevant references? |
( ) |
(x) |
( ) |
( ) |
|
Are all the cited references relevant to the research? |
(x) |
( ) |
( ) |
( ) |
|
Is the research design appropriate? |
( ) |
(x) |
( ) |
( ) |
|
Are the methods adequately described? |
(x) |
( ) |
( ) |
( ) |
|
Are the results clearly presented? |
( ) |
(x) |
( ) |
( ) |
|
Are the conclusions supported by the results? |
( ) |
(x) |
( ) |
( ) |
We hope that our revised manuscript will meet the high standard of quality needed for publication in the journal of Nanomaterials.
Thanking you for your consideration of this work.
Sincerely yours,
Xiaonan Sun
University of Paris Diderot
sun.xiaonan@univ-paris-diderot.fr
tel: 33 1 572772 13
http://www.itodys.univ-paris7.fr/fr/

Reviewer 3 Report
The manuscript entitled “2D Hetero- to Homo- Chiral Phase Transition from Dynamic Absorption of Barbituric Acid Derivatives” from Xiaonan Sun and co-workers deals with the synthesis, and hetero- to homochiral phase transition at the single-molecule level. This work extends the author’s previous work (Sun, X.; Silly, F.; Maurel, F.; Dong, C. Supramolecular chiral host-guest nanoarchitecture induced by the selective assembly of barbituric acid derivative enantiomers. Nanotechnology, 2016, 27, 42LT01) to the quite interesting topic of chiral phase transition. However, this manuscript is currently incomplete. Several points must be addressed before being considered for publication in Nanomaterials. Therefore, this manuscript should be re-submitted after being completely prepared.
1. There is no contents in the manuscript and supporting information, corresponding to the descriptions on the lines of 277 (Figure S1-2 STM results), 278 (Figure S3-9 Calculation), and 281 (DFT calculation). Therefore, this manuscript cannot be evaluated in its current state. In particular, DFT calculations are essential to reveal the intermolecular interactions upon the chiral phase transition.
2. Following terms should be correctly used: ‘adsorption’ and ‘adsorbed’, not ‘absorption’ (in the title, lines 201 and 213) and ‘absorbed’ (lines 11 and 242).
3. Reference numbers should be properly mentioned in the text. Ex. Line-93, 95, etc.
4. Chemical formula should be correct (line 91).
5. For the synthesis of 1, the authors mentioned two bases (in text NaOH, line 220; in supporting information NaH). Please correct it. K2CO3 is a mild base used in this type of reaction (ACS Omega 2022, 7, 9775–9784).
6. Reference format must be compatible according to the journal’s criteria. Some are ok. But some are different.
Author Response
Dear Editor Zhang,
Please find attached to this letter answers to the reviewers comments of our manuscript entitled “2D Hetero- to Homo- Chiral Phase Transition from Dynamic Absorption of Barbituric Acid Derivatives” that we wish to publish in Nanomaterials. Its ID is nanomaterials-2540889. The reviewer comments have been taken into account as follows:
Reviewer: 3
The manuscript entitled “2D Hetero- to Homo- Chiral Phase Transition from Dynamic Absorption of Barbituric Acid Derivatives” from Xiaonan Sun and co-workers deals with the synthesis, and hetero- to homochiral phase transition at the single-molecule level. This work extends the author’s previous work (Sun, X.; Silly, F.; Maurel, F.; Dong, C. Supramolecular chiral host-guest nanoarchitecture induced by the selective assembly of barbituric acid derivative enantiomers. Nanotechnology, 2016, 27, 42LT01) to the quite interesting topic of chiral phase transition. However, this manuscript is currently incomplete. Several points must be addressed before being considered for publication in Nanomaterials. Therefore, this manuscript should be re-submitted after being completely prepared.
- There is no contents in the manuscript and supporting information, corresponding to the descriptions on the lines of 277 (Figure S1-2 STM results), 278 (Figure S3-9 Calculation), and 281 (DFT calculation). Therefore, this manuscript cannot be evaluated in its current state. In particular, DFT calculations are essential to reveal the intermolecular interactions upon the chiral phase transition.
Author’s reply: We thank the reviewer for the comments.
We apologize that we have made a mistake. We have put wrong information for the calculation (from template of a former paper) which is misleading.We have now change the contribution description for F. M as molecular modeling where he has contributed for the plausible modeling of the different structures.
There is no DFT calculation involved in this article main text. However, we do have a discussion for future calculations concerning the phase transitions where F M will be involved.
The competition between the intermolecular hydrogen bonds and VDW interactions is a very complicated procedure. In this work, the phase transition is a display between the intermolecular hydrogen bonds, VDW interaction and molecules substrate interaction, plus the increasing of dynamic molecular adsorption. As the reviewer knows, it is quite difficult to simply clarify which kind of interaction is playing what kind of role in different structures.
Because of the large unit cell of the “nanowave” structure and the long alkyl chains, especially the dynamic transition between the nanowaves and the wire, enormous amount of work will be required for a good calculation using DFT or molecular dynamic simulations. We are having discussions to perform some calculation by the theoreticians, as it will be time consuming, we will probably write a following article mainly focused on the calculation and the interpretation of the different intermolecular competition. We wish the reviewer could kindly understand this difficulty and that the calculation is beyond the scope of this work.
On another hand, some simulation as well as experimental works concerning intermolecular hydrogen bonding and interdigitating alkyl chain vdw interaction have been reported. And we have now added new references concerning this topic, No. 45-48. .
- Following terms should be correctly used: ‘adsorption’ and ‘adsorbed’, not ‘absorption’ (in the title, lines 201 and 213) and ‘absorbed’ (lines 11 and 242).
Author’s reply: We thank the reviewer for the comments. We corrected the misspelled words in the manuscript
- Reference numbers should be properly mentioned in the text. Ex. Line-93, 95, etc.
Author’s reply: We thank the reviewer for the comments. We corrected the reference format in the paper.
- Chemical formula should be correct (line 91).
Author’s reply: We thank the reviewer for the comments. We corrected the chemical formula line 91 as C25H36N2O4
- For the synthesis of 1, the authors mentioned two bases (in text NaOH, line 220; in supporting information NaH). Please correct it. K2CO3 is a mild base used in this type of reaction (ACS Omega 2022, 7, 9775–9784).
Author’s reply: We thank the reviewer for the comment. We corrected the formula in the main text, line 229.
- Reference format must be compatible according to the journal’s criteria. Some are ok. But some are different.
Author’s reply: We thank the reviewer for the comments. We corrected all the references format in the paper. And we have reedited small mistakes in the main text.
We hope that our revised manuscript will meet the high standard of quality needed for publication in the journal of Nanomaterials.
Thanking you for your consideration of this work.
Sincerely yours,
Xiaonan Sun
University of Paris Diderot
sun.xiaonan@univ-paris-diderot.fr
tel: 33 1 572772 13
http://www.itodys.univ-paris7.fr/fr/

Round 2
Reviewer 1 Report
The authors have improved thre manuscript. However several stereochemical aspects remain to be clarified. I highlighted relevant fragments in the text and wrote remarks (pleasde, see the pdf file attached).
I recommend to pay specific attention to the Abstract of manuscript because it should seem as competently as possible to attract reader's interest to the work. Any terminological inconsistencies should be avoided.

Author Response
1. The authors have improved the manuscript. However several stereochemical aspects remain to be clarified. I highlighted relevant fragments in the text and wrote remarks (please, see the pdf file attached).
Author’s reply: We thank the reviewer for the comments and the detailed corrections.
We tried to define the sterochemical aspect , therefore the following statement was used in the main text :
«As shown in scheme S1 (see SI), the configuration on the left can then be defined as “S(p)” and that on the right as “R(p)” where p means planar. For the sake of simplicity, in what follows the two enantiomeric motifs will be referred to as TDPT-S and TDPT-R (in red and green, respectively; Figure 2b).»
We did not use the S(p) and R(p) definition throughout the whole article because we have discussed with our colleague who is an english native speaker, he strongly suggested to use the simple S and R definition because S(p) and R(p) make the english clumpy. For this reasone, we continue using the definition of S and R.
On another hand, we have followed the reviwer’s commments, we have removed the words «enantiomers» and replaced them by «enantiomeric motifs» in the main text.
Or in line 144, «TDPT-S and TDPT-R are the two motifs enantiomerically opposite»
2. I recommend to pay specific attention to the Abstract of manuscript because it should seem as competently as possible to attract reader's interest to the work. Any terminological inconsistencies should be avoided.
Author’s reply: We thank the reviewer for the comments. We have corrected the beginning of the abstract according the review’s suggestions :
« Barbituric acid derivative (TDPT) is an achiral molecule and its adsorption on a surface results in two opposite enantiomerically oriented motifs, namely TDPT- S and R. Two types of building blocks can be formed; block I is enantiomer-pure and is built up of the same motifs (format SS or RR) whereas block II is enantiomer-mixed and composes both motifs (format SR), respectively. The organization of the building blocks determines the formation of different nanoarchitectures... »
Reviewer 2 Report
Why is the author "Jean-Christophe Lacroix" omitted in the revised version? Compoud 1 is prepared using "NaH in ethanol" (line 286) while in the Suporting Info using "NaH, dry DM" (page 2). Which one is correct?
Author Response
Answers to reviewer 2:
- Why is the author "Jean-Christophe Lacroix" omitted in the revised version? Author's reply: We thank the reviewer for the comment. "Jean-Christophe Lacroix" was never one of the author in this article, not in the first verion nor in the revised version, simply because he did not participate in this work. If the reviewer has seen some remarks of removing the name of "JCL", its just because we have used the template of our last article for nanomaterials and we have made mistakes somewhere forgetting to removed the former authors. Sorry for the misunderstanding.
- Compoud 1 is prepared using "NaH in ethanol" (line 286) while in the Suporting Info using "NaH, dry DM" (page 2). Which one is correct?Author's reply: We thank the reviewer for the comment. The compound 1 is prepared using NaH in dry DMF, therefore we have now corrected the statement in the main text.
Reviewer 3 Report
The manuscript has been appropriately revised without going beyond unnecessary scope. Only a minor fix is required: In the title of the supporting information, it's 'adsorption', not 'absorption' .
Author Response
The manuscript has been appropriately revised without going beyond unnecessary scope. Only a minor fix is required: In the title of the supporting information, it's 'adsorption', not 'absorption' .
Author's reply: We thank the reviewer for the comment. The mistake has been corrected, the titles in the main manunscrip and the SI file are now :
2D Hetero- to Homo-Chiral Phase Transition from Dynamic Absorption of Barbituric Acid Derivatives